# Structural Study of Metal Binding and Coordination in Ancient Metallo-β-Lactamase PNGM-1 Variants

**DOI:** 10.3390/ijms21144926

**Published:** 2020-07-12

**Authors:** Yoon Sik Park, Tae Yeong Kim, Hyunjae Park, Jung Hun Lee, Diem Quynh Nguyen, Myoung-Ki Hong, Sang Hee Lee, Lin-Woo Kang

**Affiliations:** 1Department of Biological Sciences, Konkuk University, 120 Neungdong-ro, Gwangjin-gu, Seoul 05029, Korea; mlpiyoo@konkuk.ac.kr (Y.S.P.); sitology@konkuk.ac.kr (H.P.); diemquynh612@gmail.com (D.Q.N.); myoungkihong@gmail.com (M.-K.H.); 2National Leading Research Laboratory of Drug Resistance Proteomics, Department of Biological Sciences, Myongji University, 116 Myongjiro, Yongin, Gyeonggido 17058, Korea; kty6523@gmail.com (T.Y.K.); topmanlv@hanmail.net (J.H.L.)

**Keywords:** metallo-β-lactamase (MBL), metal coordination, antibiotics, x-ray crystallography

## Abstract

The increasing incidence of community- and hospital-acquired infections with multidrug-resistant (MDR) bacteria poses a critical threat to public health and the healthcare system. Although β-lactam antibiotics are effective against most bacterial infections, some bacteria are resistant to β-lactam antibiotics by producing β-lactamases. Among β-lactamases, metallo-β-lactamases (MBLs) are especially worrisome as only a few inhibitors have been developed against them. In MBLs, the metal ions play an important role as they coordinate a catalytic water molecule that hydrolyzes β-lactam rings. We determined the crystal structures of different variants of PNGM-1, an ancient MBL with additional tRNase Z activity. The variants were generated by site-directed mutagenesis targeting metal-coordinating residues. In PNGM-1, both zinc ions are coordinated by six coordination partners in an octahedral geometry, and the zinc-centered octahedrons share a common face. Structures of the PNGM-1 variants confirm that the substitution of a metal-coordinating residue causes the loss of metal binding and β-lactamase activity. Compared with PNGM-1, subclass B3 MBLs lack one metal-coordinating residue, leading to a shift in the metal-coordination geometry from an octahedral to tetrahedral geometry. Our results imply that a subtle change in the metal-binding site of MBLs can markedly change their metal-coordination geometry and catalytic activity.

## 1. Introduction

The increasing prevalence of multidrug-resistant (MDR) bacteria in the community and hospitals represents a worldwide concern [1,2] and poses a serious threat to the healthcare system [3]. The World Health Organization (WHO) warns that unless the necessary actions are taken against MDR bacteria, the estimated death toll from MDR bacteria could exceed that from cancer by 2050 [4].

β-lactam antibiotics, including penicillins, cephalosporins, carbapenems, and monobactams, have been the most effective antibiotics for almost 80 years since the first use of penicillin [5]. β-lactam antibiotics bind to bacterial penicillin-binding proteins (PBPs) via their four-membered β-lactam ring and interrupt the protein’s transpeptidase activity, which is required for synthesis of the peptidoglycan layer. This β-lactam-mediated inhibition of cell wall synthesis causes bacterial cell lysis. However, the emergence of β-lactamase genes and acquisition of these resistance genes by bacteria have led to a rapidly growing number of MDR bacteria that are resistant to β-lactam antibiotics [6]. β-lactamases can be divided into two groups according to their main catalytic motifs: serine β-lactamases and metallo-β-lactamases (MBLs). MBLs have a broad spectrum of hydrolytic activity and are resilient against most serine β-lactamase inhibitors; only monobactams exhibit some stability against MBLs [7].

Unlike serine β-lactamases in which a serine residue nucleophilically attacks the carbonyl carbon atom in the β-lactam ring, MBLs use a hydroxide ion to cleave the C–N bond, wherein the hydroxide ion is derived from a catalytic water molecule and coordinated by metal ion(s), usually Zn^2+^ [8,9] (Scheme 1). MBLs are classified into three subclasses: B1, B2, and B3. In general, B1 and B3 MBLs have two metal-binding sites (MBSs), whereas B2 MBLs have only one MBS [10,11]. As the substrate-attacking hydroxide ion is directly coordinated by metal ion(s) in the active site, the coordination geometry of metal binding plays an essential role in the structural and catalytic understanding of MBLs, which is critical for the development of a specific inhibitor against MBLs. The two MBSs in B1 and B3 MBLs, which are referred to as Zn1 and Zn2, have a different affinity for zinc ions [10]. B2 MBLs possess only Zn2.

PNGM-1 is derived from a metagenomic library from deep-sea sediments that predate the antibiotic era [12]. MBLs belong to the MBL superfamily, which also includes tRNase Zs [13]. PNGM-1 has both MBL and tRNase Z activities and is considered to be the evolutionary origin of B3 MBLs [14]. Like B1 and B3 MBLs, PNGM-1 has two MBSs.

MBLs hydrolyze almost all clinically used β-lactam antibiotics [15]; however, there is no effective inhibitor against MBLs yet. Compared to the catalytic serine residue in serine β-lactamases, the metal ions in the active site of MBLs do not allow the covalent binding of an irreversible or suicidal inhibitor. In addition, inhibitors with a high affinity for zinc ion(s) are toxic and not suitable for clinical applications due to putative nonspecific inhibition of ubiquitous zinc-binding proteins. We aimed to understand how metal-coordinating residues in the active site of MBLs affect the enzyme’s affinity for metal ions on a structural basis.

In this study, we determined the crystal structures of five PNGM-1 variants and confirmed that each variant exhibits a different metal binding affinity. The present structural study improves the understanding of metal binding in MBLs.

## 2. Results

### 2.1. Two Metal-Binding Sites in PNGM-1

The wild-type of PNGM-1 has two MBSs in its substrate-binding pocket (Figure 1A), and each is occupied by a zinc ion (Appendix A). Each zinc ion is coordinated by four residues and two water molecules in an octahedral geometry, whereby Asp210 is involved in the binding of both zinc ions. The zinc ion in the first MBS (Zn1) is coordinated by the residues His91, His93, His188, and Asp210; the zinc ion in the second MBS (Zn2), by Asp95, His96, His279, and Asp210. The zinc ion at Zn1 is more tightly bound with shorter coordination distances than that at Zn2 (Appendix A and Appendix A). The coordination distances between the zinc ion at Zn1 and the coordinating residues range from 2.0 to 2.9 Å, and between that at Zn2 and the residues from 2.1 to 3.0 Å. The B-factor of the zinc ion at Zn1 is lower than that of Zn2 (Appendix A).

### 2.2. Metal Coordination Geometry of the MBSs

In each MBS, a zinc ion is located at the center surrounded by six coordinating moieties, namely, four amino acid residues and two water molecules (or one hydroxide ion during the catalytic reaction) in an octahedral geometry (Figure 1B). The two octahedrons (Zn1 and Zn2) are attached to each other by sharing a plane formed by the two coordinating water molecules and Asp210. All zinc-coordinating atom distances are relatively consistent (2.0–2.3 Å) except for the long coordination distance (3.0 Å) between the zinc ions and the non-catalytic water molecule.

### 2.3. Metal Occupancy in Two MBSs

We determined the crystal structures of five PNGM-1 variants: H91A, H93A, H96A, H257A, and H279A (Figure 2 and Appendix A). His91 and His93 coordinate the zinc ion at Zn1, whereas His96 and His279 coordinate that at Zn2. His257 is not directly involved in metal coordination but plays a role in the tRNase Z activity of PNGM-1. In the structure of the H91A variant, the electron density for the zinc ion at Zn1 completely disappears compared to that in the wild-type structure, indicating the loss of the zinc ion in this variant. In the H93A variant, the electron density for the zinc ion at Zn1 is weak to allow only a partial occupancy of a zinc ion or a full occupancy for a water molecule. In the crystal structures of the H96A and H279A variants, the electron density of the zinc ion at Zn2 is markedly lower compared to that in the wild-type structure, indicating that either a water molecule with full occupancy or a zinc ion with half occupancy is present in this site. As His257 is distant from the MBSs and not involved in metal binding, the Zn1 and Zn2 in the H257A variant are occupied by metal ions.

Among all the variants, H91A and H96A exhibit the lowest metal occupancy. The MBSs in PNGM-1 are solvent exposed (Appendix A). The residues His93, Aps95, His279, and Asp210 occupy equatorial positions along the enzyme’s surface. The residues His91 and His96 are buried in the substrate-binding pocket of PNGM-1.

### 2.4. Catalytic Water Molecule between the Two Zinc Ions

The catalytic mechanism of MBLs requires a substrate-attacking hydroxide ion bound by zinc ion(s) (Scheme 1). This hydroxide ion could be directly recruited from the bulk solvent or generated by the deprotonation of a water molecule located at the same position. In the PNGM-1 structures, two water molecules are involved in metal coordination, where one (catalytic water) attacks the C–N bond of the β-lactam ring (Figure 1). In the structures of all PNGM-1 variants, except in that of H96A, the electron density corresponding to the catalytic water remains despite the complete or partial loss of a zinc ion (Figure 2).

Both water molecules in the MBSs are coordinated by four or five coordinating partners, including each other (Figure 1A). The catalytic water molecule lies on a plane formed by the four zinc-coordinating residues His93, His188, His279, and Asp95 and interacts with the non-catalytic water molecule in an apical position. The binding of the non-catalytic water with the coordination partner of the two zinc ions, Asp210, and the catalytic water is almost linear and unidirectional. The binding of the non-catalytic water molecule is weaker than that of the catalytic water. The B-factor of the catalytic water molecule is lower than that of the non-catalytic water molecule (Appendix A).

### 2.5. Structural Comparisons between PNGM-1 and Other B3 MBLs

We structurally compared the MBSs of PNGM-1 with those of tRNase Z from *Bacillus subtilis* (Bs-tRNase Z) [16] and B3 MBL from *Elizabethkingia meningoseptica* (GOB-18) [17]. In both Bs-tRNase Z and GOB-18, two metal ions are essential for enzyme activity. Although the positions of the two metal ions are well conserved among the compared enzymes, the metal coordination of Bs-tRNase Z and GOB-18 differs from that of PNGM-1. As shown for PNGM-1, Bs-tRNase Z contains two MBSs that exhibit octahedral geometries and share a common face. In contrast, GOB-18 lacks a metal-coordinating residue that corresponds to Asp210 in PNGM-1; therefore, its MBSs exhibit tetrahedral geometries share only one intersection point (Figure 3).

According to structural superimposition, the zinc positions are almost identical in the structures of PNGM-1 and Bs-tRNase Z; the distances between the zinc ions in the Zn1 and Zn2 are only 0.4 and 0.2 Å, respectively. The position of the phosphate group of the bound tRNA substrate in the Bs-tRNase Z structure superimposes well with that of the non-catalytic water molecule in the PNGM-1 structure (phosphate–water distance, 0.9 Å). This position is suitable for attack by the catalytic water molecule in PNGM-1. The catalytic water molecule is not present in the structure of the tRNA–Bs-tRNAse Z complex.

The differences in metal coordination between B3 MBL GOB-18 and PNGM-1 cause a shift in the position of the catalytic water molecule within the active site. According to structural superimposition, the position of the catalytic water molecule in the GOB-18 structure is shifted by 2.4 Å toward residue Asp210 (only present in PNGM1) compared to that in the PNGM-1 structure. The GOB-18 structure does not contain a non-catalytic water molecule.

## 3. Discussion

MBLs hydrolyze a broad spectrum of β-lactam antibiotics and are promising targets for the development of antibacterial agents, especially those against MDR bacteria. The metal ions in MBLs are essential for catalytic activity in which the catalytic water molecule, which is coordinated to the metal ions, plays a key role by attacking the β-lactam ring. We scrutinized the metal binding and coordination in wild-type PNGM-1 and in different variants with substituted residues within the MBS. Each MBS of PNGM-1 is formed by four metal-coordinating residues, and the substitution of only one of these residues dramatically reduces the metal binding affinity of the enzyme (Appendix A and Appendix A). The PNGM-1 variants lose their catalytic activity for both RNAs and β-lactam rings [14].

The metal coordination geometries in the two MBSs of PNGM-1 are almost identical. The metal coordination of Zn1 is slightly stronger than that of Zn2, as evidenced by shorter coordination distances. From a structural viewpoint, the substitution of His91 and His96 by alanine markedly weakens the metal binding affinity for Zn1 and Zn2, respectively, compared to that of the wild-type and the remaining variants. No electron density was observed for the zinc ion at Zn1 in the variant H91A. In H96A, even though electron density exists for the zinc ion at Zn2, the density is weak and noisy, implying weak binding of the metal ion. Both H91 and H96 are located deep in the catalytic pocket of PNGM-1; thus, the substitution of a bulky His residue to a sterically smaller Ala residue could lower the overall stability of the MBS, in addition to the loss of one metal coordination (Appendix A).

Geometrically, the MBS of PNGM-1 consists of two zinc-centered octahedrons sharing a common face. The common face is formed by two water molecules and Asp210. In B3 MBL GOB-18, the residue corresponding to Asp210 in PNGM-1 is missing; therefore, the MBSs of GOB-18 exhibit tetrahedral metal coordination geometries, where the two metal-centered tetrahedrons share one common point, which is the catalytic water molecule (Appendix A). The change in the metal coordination geometry shifts the position of the catalytic water molecule, altering the direction of the nucleophilic attack by the catalytic hydroxide molecule in tRNase Z and MBL.

Structural inspection of PNGM-1 structures and comparison with those of Bs-tRNase Z and B3 MBL GOB-18 showed that a variation in the metal coordination by substituting only one metal-coordinating residue could already affect the enzyme activity and substrate specificity. Currently, MBLs represent a serious threat to our community and healthcare system, and specific inhibitors against MBLs are being systematically pursued. For development of MBL inhibitors, interruption of the canonical metal coordination in MBLs could be an effective strategy. The present structural study of the metal binding and coordination in MBLs is useful for development of antibacterial agents against MDR bacteria containing MBLs.

## 4. Materials and Methods

### 4.1. Reagents

The expression vector, pET-28a(+), was purchased from Novagen (San Diego, CA, USA). The expression host, *Escherichia coli* BL21(DE3), and all restriction enzymes were purchased from New England Biolabs (Ipswich, MA, USA). Luria-Bertani (LB) medium was purchased from BD Biosciences (San Jose, CA, USA). Pre-stained protein markers for SDS-PAGE and a gel filtration calibration kit were purchased from MBI Fermentas (Hanover, MD, USA) and GE Healthcare (Piscataway, NJ, USA), respectively.

### 4.2. Gene Cloning and Site-Directed Mutagenesis

Gene cloning and site-directed mutagenesis were performed as described previously [14]. Briefly, the *PblaPNGM*-1 gene (GenBank ID: MF445022) was obtained from the functional metagenomic library from the deep-sea sediments of Edison Seamount in Papua New Guinea [18] and amplified by polymerase chain reaction (PCR) with the following primer pairs: forward primer, 5′-ATA**CCATGG**GCCACCATCATCATCATCAT*GACGACGACGACAAG*GCAGGTGGAAAAGTAACCTC-3′; backward primer, 5′-GAG**AAGCTT**TTAGCTTCCCACTCCCAAATC-3′ (restriction sites are written in bold, the hexahistidine (His6)-tag site is underlined, and the enterokinase (EK) recognition site is highlighted in italics). The amplified DNA and pET-28a (+) vector (Novagen, San Diego, CA, USA) were double-digested with NcoI and HindIII, and the digested DNA was then cloned into the digested pET-28a (+) vector.

Site-directed mutagenesis to generate the PNGM-1 variants H91A, H93A, H96A, H257A, and H279A was performed using a QuikChange II Site-Directed Mutagenesis Kit (Stratagene, Agilent Technologies, Santa Clara, CA, USA) according to the manufacturer’s instructions. After verifying the DNA sequences, the resulting plasmids expressing the corresponding PNGM-1 variants were individually transformed into *E. coli* BL21(DE3) cells.

### 4.3. Expression and Purification of PNGM-1 Variants

The transformed *E. coli* BL21(DE3) cells containing the pET-28a (+) vectors with the PNGM-1 variant genes were grown in LB medium containing 50 mg/mL kanamycin at 303 K until an OD_600nm_ of 0.6 was reached. Then, 1 mM isopropyl β-D-1-thiogalactopyranoside (IPTG) was added to the culture to induce protein expression. After 16 h of cultivation at 289 K, the cells were harvested by centrifugation at 4700× *g* for 10 min at 277 K and resuspended in ice-cold 20 mM sodium phosphate pH 7.0. The cells were disrupted by sonication and centrifuged at 20,000× *g* for 60 min at 277 K. The clarified supernatant was loaded onto a His-Bind column (Novagen, San Diego, CA, USA) equilibrated with binding buffer (20 mM sodium phosphate pH 7.9, 10 mM imidazole, 500 mM NaCl). The His-tagged PNGM-1 variants were eluted with the same buffer containing 500 mM imidazole. For further purification, the His6-tag was removed from the PNGM-1 variants using enterokinase according to the manufacturer’s instructions (Novagen). The reaction mixture was desalted and concentrated using a Fast Desalting column (Amersham Biosciences, Buckinghamshire, UK) and then loaded onto a Mono S column (Amersham Biosciences, Buckinghamshire, UK) pre-equilibrated with 10 mM sodium phosphate pH 7.0. PNGM-1 variants were eluted with a linear gradient of NaCl (0–0.5 M) in phosphate buffer. The proteins were further purified by size-exclusion chromatography on a Superdex 200 (16/60) column (GE Healthcare) equilibrated with 10 mM MES, pH 6.8, at a flow rate of 1 mL/min.

### 4.4. Crystallization

The purified wild-type PNGM-1 and its variants H91A, H93A, H96A, H257A, and H279A were crystallized in 96-well Intelli-Plates (Art Robbins) and Hampton research buffer sets at 287 K using a Hydra II e-drop automated pipetting system (Matrix). Crystallization drops consisted of 0.5 µL protein solution and 0.5 µL reservoir solution, which were equilibrated against 50 μL reservoir solution at 287 K. After obtaining initial crystals, we optimized the crystallization conditions and refined it in terms of reliability. Each crystallization condition was as described in Appendix A. For crystal optimization in 24 wells plates, hanging drops were prepared by mixing 1 μL protein solution with the same volume of mother liquor. Fully-grown single crystals were transferred to a mixture of mother liquor and cryoprotectant solution (20% glycerol, 20% ethylene glycol, or 15% MPD). Crystals were flash-cooled in liquid nitrogen at 100 K for data collection.

### 4.5. Data Collection, Structure Determination, and Refinement

X-ray data were collected at the beamlines BL-5C and BL-11C at the Pohang Light Source in South Korea. Data were integrated and scaled with the DENZO and SCALEPACK algorithms [19]. Initial phases of the structures were determined by molecular replacement (MR) with the phaser program of the CCP4 software package [20]. The structure of PNGM-1 (PDB ID: 6j4n) was used as the search model for MR. Manual model building into the electron density map was performed using the COOT program [21]. Multiple cycles of structure refinement were performed with Refmac5 of the CCP4 software package [22]. The coordinates and geometries of all structures were validated by WHATIF [23] and SFCheck [24]. Graphic presentations were created with PyMOL [25]. The diffraction data and structure refinement statistics are provided in Table 1.

Values in parentheses are for the shell with the highest resolution.
(1)$$R{\mathrm{merge}}_{=}\sum_{hkl}\sum_{i\;}\left|\left(I_{i}\left(hkl\right)\right)-I\left(hkl\right)\right|/\sum_{hkl}\sum_{i}I_{i}\left(hkl\right)\;$$
where $I_{i}\left(hkl\right)$ is the mean intensity of the *i*th observation of symmetry-related reflections *hkl*.
(2)$$R_{\mathrm{work}}=\sum_{hkl}\left|\left|F_{obs}\right|-\left|F_{calc}\right|\right|/\sum_{hkl}\left|F_{obs}\right|$$
where $F_{calc}$ is the calculated protein structure factor from the atomic model (*R*_free_ was calculated as *R*_work_ with a randomly selected 5% of the reflections).

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
