# Peer review of "Structural Study of Metal Binding and Coordination in Ancient Metallo-β-Lactamase PNGM-1 Variants"

_ijms, 2020, doi:10.3390/ijms21144926_

Round 1

Reviewer 1 Report

I would like to congratulate the authors for the descriptive work. This manuscript is an interesting structural study of the metal binding and coordination of the ancient metallo-β-lactamase PNGM-1 and substitution variants. The manuscript is pleasantly short and yet precisely written.

However, a few questions remain open and unresolved:

First, what happens with the zinc coordination in the MBS (Zn1 & Zn2) by substitution of Asp210? Is this variant available?

At two points the authors consider a reduced metal binding affinity of the substitution variants:

Line 108-109: “…residues, the substitution of even a residue significantly reduces the affinity for metal binding at the corresponding MBS.”

Line 173 & 174: “Each MBS of PNGM-1 is formed by four residues and the substitution 173 of even a residue dramatically lowered the metal binding affinity at the corresponding MBS.”

Here the appropriate methodology for the statement should be chosen and used. The metal binding affinities of the MBS could be determined by ITC in comparison of WT protein and substitution variants.

At two other points, the limitations of the methodology used for detecting zinc ions in MBS are addressed:

Line 110-113: “In PNGM-1 H93A, the electron density for Zn1 is weak to allow only a partial occupancy of a zinc ion or a full occupancy for a water molecule. For Zn2, both PNGM-1 H96A and H279A only weakly maintained the electron density just for a water molecule or a zinc ion with a half occupancy.”

Line 180 & Line 181: “In H96A, even though the density for Zn2 still exists, the density is not good and noisy which implies the weak binding of metal ion.”

In order to verify the results, a quantitative determination of the zinc cofactor in molar ratio to the protein variants is helpful, possibly using an element determination by AAS or ICP-MS.

Reviewer 2 Report

The authors describe an accurate structural analysis of 5 mutants of the PNGM-1 enzyme, produced by site directed mutagenesis to identify the strength of histidine residues in coordinating zinc ions. They call PNGM-1 an ancient metallo-β-lactamase rather then an ancestor of this class of enzymes, as they discussed in their previous work where they described the activity of such enzyme. Structural data are correlated to activity, again already published in the previous paper, while only structural analysis is presented here.

A more estensive comparison with known metal dependent b-latamases would enrich this paper, despite part of these analysis has already been done in their previous publication.

Since main analysis and discussion is based on Zn coordination, I suggest to produce figures where not only 2Fo minus Fc density is shown but also omit map for Zn atoms are evidenced, since any level of occupancy and or absence is definitely better demonstrated by these types of experimental calculated maps.

Given the wavelength they used to collect data, Zn atoms can be observed even by anomalous maps, that would be even better to be calculated and presented here, at least for the best substituted metal sites (maybe in the suppl. data).

I think authors should try some other techniques in solution at least to quantify Zn amount bound to each of the 5 PNGM-1 mutants, to corroborate and strengthen structural data.

Crystallographic data are of good quality and refined structures present good parameters. Given the very specific type of analysis, deeply focused on mutants structural analysis, I was wondering if a more dedicated journal would be more appropriate. 

I strongly suggest an extensive editing of English language and style.

Minor revisions:

From what concern Zn occupancy, I suggest to include B factors comparison tables in the main text, if possible, or at least average B factors values in the text of the manuscript, for more accurate description of atomic thermal displacement.

Reviewer 3 Report

Indeed, metallo-β-lactamase (MBLs) production is a concerning β-lactam resistance mechanism contributing to Multi-Drug Resistant (MDR) strain development that constitute a serious clinical threat to public health system. The manuscript is exhaustive, clear and well written, providing a valuable tool to understand the binding and coordination of metal ions to help in developing MBL inhibitors, suitable for treat infections caused by pathogens producing those MBLs. Methodology is well explained and plenty of detail, and Supplementary material included is exhaustive and very illustrative.

I just have some minor issues to be adressed:

Line 46 – β-Lactamase instead of B-Lactamase

Section 4.2 – Restriction site in primers should be in bold, as well as hexahistidine (His6) tag site should be underlined, and enterokinase (EK) recognition site should be in italic, as indicated in text.

Author Response

Indeed, metallo-β-lactamase (MBLs) production is a concerning β-lactam resistance mechanism contributing to Multi-Drug Resistant (MDR) strain development that constitute a serious clinical threat to public health system. The manuscript is exhaustive, clear and well written, providing a valuable tool to understand the binding and coordination of metal ions to help in developing MBL inhibitors, suitable for treat infections caused by pathogens producing those MBLs. Methodology is well explained and plenty of detail, and Supplementary material included is exhaustive and very illustrative.

I just have some minor issues to be adressed:

Line 46 – β-Lactamase instead of B-Lactamase

>> The typo is fixed.

 “B-Lactamase” to “β-Lactamase”

Section 4.2 – Restriction site in primers should be in bold, as well as hexahistidine (His6) tag site should be underlined, and enterokinase (EK) recognition site should be in italic, as indicated in text.

>> The manuscript is fixed as mentioned.

Line 229-233: “… forward primer, 5’- ATACCATGGGCCACCATCATCATCATCATGACGACGACGACAAGGCAGGTGGAAAAGTAACCTC-3’; backward primer, 5’-GAGAAGCTTTTAGCTTCCCACTCCCAAATC-3’ (restriction sites are written in bold, the hexahistidine (His6)-tag site is underlined, …”

Round 2

Reviewer 1 Report

The authors' answers have answered the questions in an acceptable way.

Author Response

The reviewer 1 accepted the 1st round revised manuscript.

Reviewer 2 Report

The authors introduced anomalous maps in the supplementary data, unfortunately only for the wt protein, but at least for that the occupancy and presence of metal ions is well supported. 

FoFc maps sounds not properly called...are those maps omit or difference maps? I suppose you are showing omit map since I don't expect such an Fo minus Fc residual map after refinement, unless wrong occupancy has been attributed to metal atoms. Please clarify the nature of the calculated maps in the legends and any text where you refer to them. 

The authors haven't produced any data from a second independent technique to determine the amount of Zn in the wt and mutated variants, that would strengthen the results obtained, but simply discussed the difficulties in properly estimating Zn specifically bound to the site of interest. As a matter of fact also crystallographic data could suffer from the effect of crystallization conditions and so on, which could interfere with the chelating properties of the active site.

Nevertheless, I recommend to publish this paper, after minor adjustments.

Author Response

>>> The FoFc map is corrected as the omit map.

Fig. S3. The metal binding site of PNGM-1 wild-type and variants with the omit map: (a) PNGM-1 wild-type, (b) H91A variant, (c) H93A variant, (d) H96A variant, (e) H257A variant, and (f) H279A variant. The omit map of metal ions and water molecules is contoured at 4.0 e/Å3 except the that of H93A variant (contoured at 10.0 e/Å3).